# Age Estimation in Brazilian Adults Using the Pulp/Tooth Ratio of the Maxillary Canine and Mandibular Second Premolar

**DOI:** 10.3390/diagnostics14070749

**Published:** 2024-03-31

**Authors:** Ismar Nery-Neto, Orlando Aguirre Guedes, Lucas Rodrigues de Araújo Estrela, Luciano Tavares Angelo Cintra, Cyntia Rodrigues de Araújo Estrela, Carlos Estrela

**Affiliations:** 1Department of Endodontics, School of Dentistry, Federal University of Goiás, Goiânia 74605-020, Brazil; estrela3@terra.com.br; 2Department of Oral Biology, School of Dentistry, Evangelical University of Goiás, Anápolis 75083-515, Brazil; orlandoaguedes@gmail.com (O.A.G.); estrelacyntia@gmail.com (C.R.d.A.E.); 3Department of Preventive and Restorative Dentistry, School of Dentistry, São Paulo State University, Araçatuba 16015-050, Brazil; estrelalucas4@gmail.com (L.R.d.A.E.); lucianocintra@hotmail.com (L.T.A.C.)

**Keywords:** human identification, age estimation by teeth, forensic dental radiography, forensic science, forensic dentistry

## Abstract

(1) Background/Objectives: Accurate determination of chronological age is crucial in legal dental identification. This study aimed to compare the effectiveness of different formulas in estimating the age of a Brazilian subpopulation by analyzing the pulp/tooth ratio of the maxillary canine and mandibular second premolar in panoramic and periapical radiographs. (2) Methods: The sample consisted of panoramic and periapical radiographs of 247 individuals. The file of each radiograph was opened in the Adobe Photoshop CS4^®^ program to outline and obtain values in pixels for calculating the pulp/tooth ratio. Statistical analysis was conducted using the SPSS program, with a significance level set at 5%. (3) Results: The interclass correlation coefficient demonstrated excellent intra-observer agreement (0.990–0.999). The determination coefficients (R^2^) suggested that only 30–35% of the actual age results could be explained by the pulp/tooth ratio. The smallest differences were observed with Cameriere’s formula for the mandibular second premolar on panoramic radiographs (+4.1 years). The greatest differences were found with the formulas for the mandibular second premolar in panoramic radiographs of the Korean (+12.5 years) and Portuguese (−12.1 years) populations. (4) Conclusions: The equations employed showed little agreement between the actual age and the estimated age.

## 1. Introduction

The identification of individuals through the dental arch is commonly employed in disaster victim cases and forensic scenarios to ascertain dates of birth and death [1,2]. Accurate determination of the chronological age is crucial in legal dental identification, cases requiring proof of a subject being underage, matters related to alimony, retirement claims [3], criminal majority, and public policies for individuals with no personal documentation [4,5,6,7,8].

Global migration and the settlement of refugees worldwide have increased the need for physicians and forensic dentists to determine the chronological age of individuals using biological age estimation techniques [9,10]. Radiographic examinations, being readily available, non-invasive, and reproducible, are extensively used for this purpose [1,4,6,10,11,12,13,14,15,16,17,18].

Age estimation in the adult population represents a real challenge, given that the changes in teeth and supporting structures are more subtle compared to the stages of tooth formation and eruption [7,19]. The forensic literature generally accepts a margin of error between 6 and 8 years [20], preferably less than 10 years [7]. Current recommended methods involve evaluating the continuous process of dental mineralization that occurs with advancing age [17,21] through the calculation of the pulp/tooth ratio on radiographs [5,19,22,23,24,25,26,27] or linear measurements of pulp/root length; length of pulp/tooth; tooth length/root; and pulp/root width in three levels [7,28,29,30].

Given Brazil’s mixed population [13], the relationships and variations in the pulp/tooth ratio may differ significantly when compared to more ethnically uniform populations [10,18,30,31,32,33]. The present study aimed to compare the effectiveness of different equations in estimating the age of an adult Brazilian subpopulation by calculating the pulp/tooth ratio of the maxillary canine and mandibular second premolar in panoramic and periapical radiographs, and to develop equations for estimating age based on values obtained from this sample. The null hypothesis tested was that there would be no significant difference in the correlation between actual and estimated ages using the different formulas.

## 2. Materials and Methods

This study underwent thorough review and received approval from the Research Ethics Committee of the Evangelical University of Goiás, Brazil (CAAE 53107721.0.0000.5076).

With a 95% confidence interval and 90% power to achieve correlation confidence of 0.3 between chronological age and pulp/ratio, a minimum of 100 samples were estimated for each type of tooth.

This study included 247 individuals (494 teeth) of both genders, ranging in age from 20 to 82 years (Table 1). Panoramic and periapical radiographs from the database of a dental imaging diagnostic center in Goiânia-GO, Brazil, were utilized for assessment.

Inclusion criteria comprised the presence of at least one healthy maxillary canine and one mandibular second premolar on both the right and left sides, without endodontic treatment, crowns, and prosthetic posts. Additionally, participants had to exhibit an absence of calcification in the root canal, external or internal root resorption, fully formed apex, no history of orthodontic treatment, and no developmental disorders and pathological processes. Images with blurriness, distortions, low density, and poor contrast were excluded from this study.

The points were marked, and the dental structures were delineated using the polygonal lasso tool of the Adobe Photoshop CS4^®^ program (Adobe Systems Inc., San Jose, CA, USA). A high-resolution Samsung^®^ monitor, model S24E310 (Samsung, Manaus, Brazil), was used for image visualization. All images were outlined using a pen and Huon^®^ Inspiroy 420 graphics tablet (Shenzhen Huion Animation Technology Co., Ltd., Shenzhen, China). The teeth were classified according to the FDI World Dental Federation notation [32]. Initially, teeth 13 and 45 were evaluated. In the absence of either of these, teeth 23 and 35 were evaluated. Measurements were taken on the maxillary canine and mandibular second premolar on panoramic radiographs, followed by measurements on periapical radiographs. At least 20 points outlining the contour of the tooth and at least 10 points delineating the contour of the pulp were marked (Figure 1 and Figure 2). This process determined the total area in pixels of the tooth and pulp [2,5,11,12,13].

All data were recorded in a Microsoft Excel^®^ spreadsheet (Microsoft Corporation, One Microsoft Way, Redmond, WA, USA) and later exported to the IBM^®^ SPSS program version 20.0 (IBM Corporation, Armonk, NY, USA).

Age estimates were conducted by applying the pulp/tooth ratio values obtained from maxillary canines and mandibular second premolars on panoramic and periapical radiographs, utilizing specific formulas [2,5,11,12,13] (Table 2).

Linear regression was used to assess the correlation between the pulp/tooth ratio values in pixels of the maxillary canines and mandibular second premolars and the age of the participants in the sample. Additionally, specific equations were constructed for the Brazilian adult subpopulation based on these findings.

All measurements were conducted by a single observer, a dental radiology specialist with 30 years of professional experience, who underwent prior calibration. To assess intra-observer reproducibility, a random sample of 50 radiographs (25 panoramic and 25 periapical) was measured after a 30-day interval.

The normality of quantitative variables was evaluated using the Kolmogorov–Smirnov test. The intraclass correlation coefficient (ICC) was used to assess intra-observer agreement. The values were interpreted using the Domeinc Cicchetti classification: values <0.4 = poor agreement, 0.4 to 0.6 = fair agreement, 0.6 to 0.75 = good agreement, and 0.75 to 1.0 = excellent agreement [34]. Categorical variables were described by frequencies. Quantitative variables with normal distribution were presented as mean and standard deviation, while those with asymmetrical distribution were described using median and interquartile range (25th and 75th percentiles). Quantitative variables with normal distribution were compared using Student’s *t*-test for paired samples and asymmetric variables using the Wilcoxon test. The Bland and Altman technique was used to assess the agreement between the ages calculated by the different formulas and the actual age. A significance level of 5% was considered for all analyses.

## 3. Results

The ICC data are presented as the median values of the measurements conducted at different times. All measurements exhibit ICC values exceeding 0.99% (Table 3).

Table 4 presents a comparison of the estimated age of the Brazilian adult subpopulation by calculating the pulp/tooth ratio of the maxillary canine using the equations developed by Cameriere et al. [5], Cameriere et al. [2], Azevedo et al. [11], Lee et al. [12], and Anastácio et al. [13]. Table 5, on the other hand, focuses on equations applied to mandibular premolars.

The results of the *t*-test indicated a low statistical correlation. Additionally, the Bland and Altman test with a 95% confidence interval revealed that the values, both positive and negative, were highly dispersed and distant from the mean, indicating minimal agreement between the data. The smallest differences were observed with Cameriere’s formula [2] for the mandibular second premolar on panoramic radiographs (+4.1 years) (Table 5). The greatest differences were found with the formulas for the mandibular second premolar in panoramic radiographs of the Korean [12] (+12.5 years) and Portuguese [13] (−12.1 years) populations.

The average values of the pulp/tooth ratio for the maxillary canines in the panoramic and periapical radiographs were 0.1025 and 0.1064, respectively (Table 6). For the mandibular premolars, the average values were 0.0992 in panoramic radiographs and 0.0937 in periapical radiographs (Table 6).

The representations of the variables in this study, represented by the pulp/tooth ratio and the actual age, are illustrated in the form of boxplot charts for the maxillary canines and mandibular second premolars in both panoramic and periapical radiographs Figure 3). Additionally, the data were organized in dispersion diagrams, which show the relationship between age and the pulp/tooth ratio, with a separate analysis for each gender (Figure 4).

The boxplot charts of the pulp/tooth ratio of the maxillary canines on panoramic radiographs (Figure 3A) illustrate medians with positive and negative asymmetrical distributions, particularly accentuated between 60 and 69 years of age. High interquartile ranges indicate similar pulp/tooth ratio values in over 50% of the cases among the 30–39, 40–49, and 50–59 age groups.

The boxplot charts of the pulp/tooth ratio of the maxillary canines on periapical radiographs (Figure 3C) illustrate medians with positive and negative asymmetrical distributions, which are particularly noticeable between the ages of 20–29, 50–59, and 70 years or older. High interquartile ranges indicate similar pulp/tooth ratio values in over 50% of the cases among the 30–39, 40–49, 50–59, and 70 years or older age groups, as well as between the 60–69 and 70 years or older age groups.

The boxplot charts of the pulp/tooth ratio of the pulp/tooth ratio of the mandibular second premolars on the panoramic radiographs (Figure 3B) illustrate medians with positive and negative asymmetrical distributions, which are particularly noticeable between the ages of 30–39 and 60–69 years. High interquartile ranges indicate similar pulp/tooth ratio values in over 50% of the cases among the 30–39, 40–49, 50–59, and between the 60–69 and 70 years or older age groups.

The boxplot charts of the pulp/tooth ratio of the mandibular second premolars on the periapical radiographs (Figure 3D) illustrate medians with positive and negative asymmetrical distributions, which are particularly noticeable between the ages of 40–49, 60–69, and 70 years or older. High interquartile ranges indicate similar pulp/tooth ratio values in over 50% of the cases among the 30–39, 40–49, 50–59, and between the 60–69 and 70 years or older age groups.

The dispersion diagrams of the pulp/tooth ratio of the maxillary canines and mandibular second premolars on panoramic and periapical radiographs (Figure 4) did not demonstrate any discernable linear or non-linear relationship.

Table 7 presents the equations derived from linear regression analysis between the pixel values of the pulp/tooth ratio of the maxillary canines and mandibular second premolars and the ages of the sampled individuals in the Brazilian subpopulation. These linear regressions yielded four specific formulas applicable to maxillary canines and mandibular second premolars in both panoramic and periapical radiographs. The determination coefficients (R^2^) revealed that only 30 to 35% of the actual age could be explained by the pulp/tooth ratio. The average age of the sampled Brazilian subpopulation was 36.3 years, with an SD of ±12.7 years, indicating a dispersion primarily between 23.6 and 49 years.

## 4. Discussion

The present study aimed to compare the effectiveness of different formulas in estimating the age of a Brazilian subpopulation by calculating pulp/tooth ratio values in maxillary canines and mandibular second premolars using panoramic and periapical radiographs. The results revealed minimal agreement between the actual and estimated ages using the equations developed by Cameriere et al. [5], Cameriere et al. [2], Azevedo et al. [11], Lee et al. [12], and Anastácio et al. [13]. Thus, the tested null hypothesis was rejected.

Among the conventional radiographic methods for estimating age in adults, those proposed by Cameriere [1,2] and Kvaal [3] are the most widely studied and utilized [4]. The Cameriere method was chosen for this experiment due to its ease of reproducibility and the proportional nature of the obtained values [2,5], as opposed to the metric approach of Kvaal’s method [3,6,7]. Additionally, Cameriere’s method is less prone to geometric errors commonly found in radiographic images [8,9].

Previous studies employing the pulp/tooth ratio strategy typically selected single-rooted teeth, as they undergo fewer morphological changes over time. Among the dental groups investigated, maxillary canines and mandibular premolars were the most used [2,5,7,10,11,12,13,14,15]. It is noteworthy that Cameriere’s methodology [5] recommends using a minimum of 20 points for outlining the tooth and 10 points for outlining the pulp. For the execution of this study, a pen and digitizing table were chosen to ensure ease, precision, and standardization of markings [16]. The consistently high ICC values exceeding 0.99% across all measurements underscore the reproducibility and standardization achieved in this study.

In this study, the formulas utilizing the pulp/tooth ratio of canines showed the smallest difference between actual and estimated age when using the equation proposed by Cameriere et al. [5] with periapical radiographs (+7 years). Although originally developed for application in periapical radiographs, its application in panoramic radiographs resulted in a difference of +9.1 years. Interestingly, this value closely aligned with the difference obtained (+8.9 years) using a specific formula for the Brazilian population with periapical radiographs [11].

For lower premolars, the smallest difference (+4.1 years) was observed on panoramic radiographs after applying the equation by Cameriere et al. [2]. While this formula was initially developed for panoramic radiographs, its application to periapical radiographs yielded a difference of +6.1 years. Notably, this value closely resembled the actual age when compared to the values obtained using specific formulas for Korean [12] and Portuguese [13] populations produced in panoramic radiographs.

Although the original formulas developed for maxillary canines and mandibular premolars [2,5] yielded lower values when compared to other formulas [11,12,13], all results demonstrated statistically significant differences (*p* < 0.001). In the case of maxillary canines, the estimated ages surpassed the actual ages. The standard deviation (SD) ranged from ±10 [11] to ±12.6 years [2], indicating considerable variability in the data. Additionally, the confidence interval (CI) exhibited wide values, signifying limited agreement.

Similarly, the estimated ages for mandibular second premolars also exceeded the actual ages, except when using the formula for the Portuguese population [13]. The SD ranged from ±1.8 [13] to ±10.5 years [12]. Like the canines, the CI displayed large values, indicating limited agreement, despite the forensic literature accepting a margin of error between 6 to 8 years [17], and preferably less than 10 years [4].

Brazil presents a mixed population [13], indicating that the relationships and variations in the pulp/tooth ratio may differ significantly when compared to more ethnically uniform populations [10,18,30,31,32,33]. This underscores the potential need for the development of specific population formulas. The measurements obtained in this study underwent linear regressions, resulting in the construction of four equations tailored for each type of radiograph (panoramic and periapical) (Table 7). This approach is warranted due to specific geometric variations inherent in panoramic and periapical radiographs [8,9,18]. Previous studies have often applied the same formula to both panoramic and periapical radiographs [15,19].

The number of exams/patients included in the studies exhibits considerable variation. The smallest sample size observed was 30 individuals [20], while the largest included 606 individuals [2]. Challenges with sample sizes are inherent in studies aimed at determining age in adults, particularly after applying stringent inclusion and exclusion criteria [6,17]. Similar challenges were encountered in the present study. Research in this field suggests incorporating formulas derived from diverse populations or augmenting the sample size [7,10,15,21,22,23,24].

The absence of methodological standardization impedes or complicates the establishment of correlations and hinders the explanation of certain results [4]. Some studies express data in terms of standard error (SE) [5,12,25], while others use SD [11,26]. Additionally, some studies provide the R^2^ values [15,27] while simultaneously reporting values for EP and mean error (EM) [13]. In contrast, this research, alongside R^2^, presented its results in terms of SD, mean absolute error (EMA), and EP. This approach facilitated comparisons with the original formulas [4].

The linear regression analysis of maxillary canines on periapical radiographs revealed R^2^ = 0.35 and EP = 10.26. These values are notably discrepant and higher when compared to the original formula [5], which reported R^2^ = 0.86 and EP = 5.45. Similarly, the linear regression values for mandibular premolars on panoramic radiographs showed R^2^ = 0.32 and EP = 10.47, whereas the original formula indicated R^2^ = 0.75 and EP = 7.42 [2].

The extreme quartile diagrams suggest a direct correlation between increasing age and secondary dentin deposition, as evidenced by the gradual decrease in pulp/tooth ratio values among individuals aged 20 to 70 years or older. However, these diagrams also exhibit wide ranges and lack concentration of data points, indicating a weak correlation between age group and pulp/tooth ratio. In essence, individuals across different age groups exhibit similar values, aligning with findings for mandibular second premolars in the Portuguese population [13], but diverging from results reported in other studies [1,2,5,15], where this correlation appeared stronger.

The scatter plots did not reveal a linear correlation, consistent with the findings from the extreme quartile plots. This observation suggests the absence of sexual dimorphism, aligning with previous reports in the literature [2,5,11,12,13].

The estimation of the age in living individuals requires higher accuracy compared to age estimation in cadavers or skeletons [26]. However, the range of applicable methods is limited [17], categorized into morphophysiological, radiographic, and biomechanical [28], each with its indications and specificities [29,30]. Among the non-invasive methods, imaging tests are widely employed due to their ease, reproducibility, and applicability in both in vivo and ex vivo settings [6,28,31,32]. It is evident that imaging tests utilized in children and adolescents, which classify or measure during the mineralization, formation, development, and eruption phases of the teeth present, yield more precise results compared to the methods used in adults, even when applying foreign formulas [33,34,35,36,37,38].

Periapical radiographs can exhibit distortions, particularly in the vertical angle, resulting in elongated or shortened images, despite the use of radiographic positioners [8]. Panoramic radiographs, in addition to magnification, may also show vertical and horizontal distortion [9] inherent to each device, individual anatomical variations, or arising from minor positioning errors [8,18,39,40,41,42,43,44]. Another limitation of conventional exams is the projection of two-dimensional images of three-dimensional structures [45], restricting any analysis, quantitative or qualitative, to the vertical and mesiodistal dimensions of teeth or anatomical structures [46]. The morphology of the root canal can change over time, with secondary dentin deposition occurring with aging [47,48]. Young patients exhibit greater variations compared to older patients [49]. Inclusion specialists in Dental Radiology in the research team could enhance sample selection and measurement execution [4].

It is important to acknowledge certain limitations of this study. Firstly, the formulas were developed for adult populations; therefore, age estimation in elderly individuals and adolescents/young adults may yield less precise results. Secondly, this study utilized a database from only one dental imaging diagnostic center, which may not fully represent the geographic variation in dental characteristics and their relationship with age. This limitation could result in less accurate age estimates in populations with distinct dental characteristics due to environmental or genetic factors. Thirdly, this study was limited to the maxillary canines and mandibular second premolars. In conditions where these teeth are missing, the method cannot be employed.

Cone beam computed tomography represents a new technology with increased significance across various dental disciplines. Numerous studies investigating age estimation have emerged utilizing this tool [48,50,51]. Despite promising prospects [47], challenges akin to those encountered in conventional methods have been noted [50,51,52,53,54,55]. Human programming and artificial intelligence stand as potential techniques to mitigate sample selection and evaluation biases in the future [56,57,58]. New investigations should be encouraged with larger sample sizes and the development of specific population formulas. It is pertinent for forensic dentistry that future studies compare the results obtained through different imaging modalities.

## 5. Conclusions

There was little agreement between the actual age and the estimated age when using the equations. New equations were developed specifically for estimating the age of adult individuals using maxillary canines and mandibular second premolars in both panoramic and periapical radiographs.

## Figures and Tables

**Figure 1 diagnostics-14-00749-f001:**
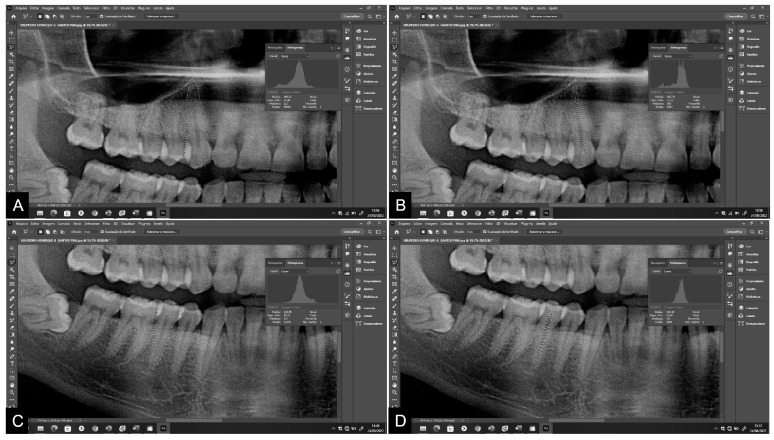
Measurements of tooth and pulp areas and pixel values on panoramic radiograph: (**A**,**B**) maxillary canine; (**C**,**D**) mandibular second premolar.

**Figure 2 diagnostics-14-00749-f002:**
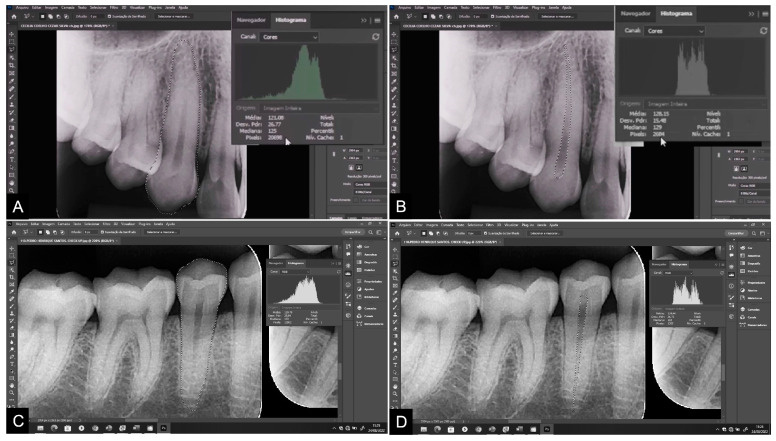
Measurements of tooth and pulp areas and pixel values on periapical radiograph: (**A**,**B**) maxillary canine; (**C**,**D**) mandibular second premolar.

**Figure 3 diagnostics-14-00749-f003:**
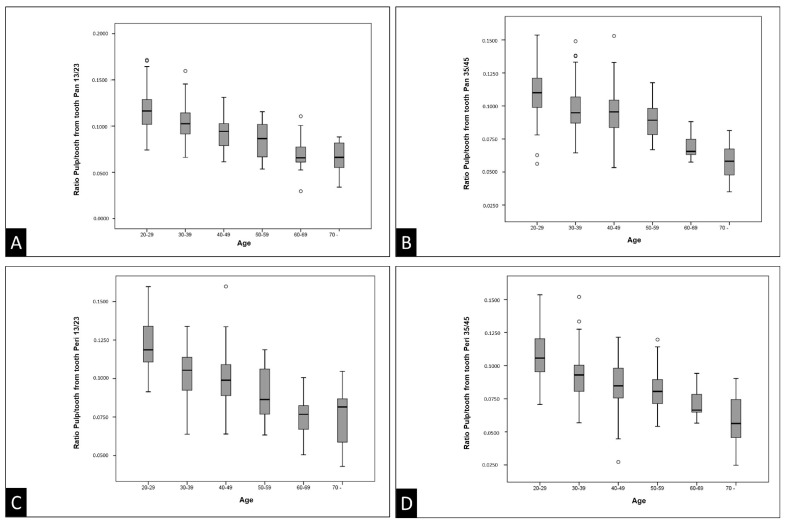
Boxplot charts of the pulp/tooth ratio of maxillary canines (**A**,**C**) and mandibular second premolars (**B**,**D**) on panoramic and periapical radiographs. Pan: panoramic radiography. Peri: periapical radiography.

**Figure 4 diagnostics-14-00749-f004:**
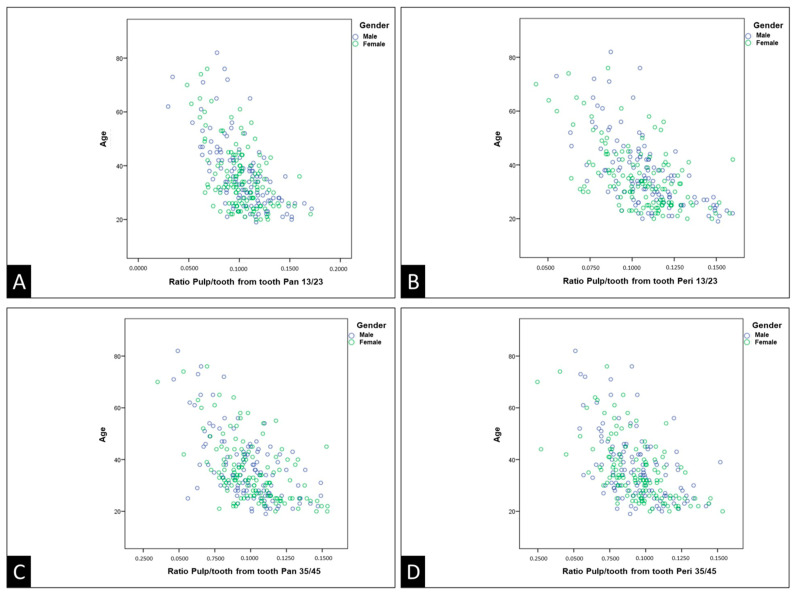
Dispersion diagrams of the pulp/tooth ratio of de maxillary canines (**A**,**B**) and mandibular second premolars (**C**,**D**) on panoramic and periapical radiographs. Pan: panoramic radiography. Peri: periapical radiography.

**Table 1 diagnostics-14-00749-t001:** Distribution of sample by age and gender.

Age	Male	Female	Total
20–29 years old	40	44	84
30–39 years old	38	46	84
40–49 years old	22	22	44
50–59 years old	8	10	18
60–69 years old	4	5	9
70 years old or older	5	3	8
Total	117	130	247

**Table 2 diagnostics-14-00749-t002:** Studies, radiographic examinations, and applied equations for age estimation.

Author	Radiographic Exam	Equation
Cameriere et al. [5]	Periapical	A.E = 99.973 − 532.775 × R13
Cameriere et al. [2]	Panoramic	A.E = 76.29 − 362.57 × R45
Azevedo et al. [11]	Periapical	A.E = 94.706 − 465.358 × R13
Lee et al. [12]	Panoramic	A.E = 97.083 − 487.415 × R45
Anastácio et al. [13]	Panoramic	A.E = 32.493 − 83.890 × R45

A.E: age estimate. R: pulp/tooth ratio.

**Table 3 diagnostics-14-00749-t003:** ICC for inter-observer assessment of pixel measurements.

	Measure 1	Measure 2	*p* *	ICC (CI 95%)
Pan T13/23				
Tooth area	19,013	18,762	0.579	0.999
Pulp area	2110	2081	0.606	0.997
Peri T13/23				
Tooth area	22,501	22,686	0.643	0.990
Pulp area	2523	2552	0.239	0.996
Pan T35/45				
Tooth area	15,139	15,610	0.717	0.999
Pulp area	1623	1619	0.116	0.998
Peri T35/45				
Tooth area	15,656	15,540	0.055	0.999
Pulp area	1534	1449	0.557	0.995

Data presented by the median. * Wilcoxon test. ICC: interclass correlation coefficient. CI: confidence interval. Pan: panoramic radiography. T13/23: tooth 13/23. Peri: periapical radiography. T35/45: tooth 35/45.

**Table 4 diagnostics-14-00749-t004:** Comparison of actual age and estimated age by various equations for canines (*n* = 247).

Equation	Mean ± SD (Years)	Difference in Years	*p* *	CI (95%) **
Cameriere et al. [5]				
Pan T13/23	45.4 ± 12.6	+9.1	<0.001	−13.4 to 31.6
Peri T13/23	43.3 ± 11.4	+7.0	<0.001	−14.5 to 28.5
Azevedo et al. [11]				
Peri T13/23	45.2 ± 10.0	+8.9	<0.001	−11.7 to 29.5

SD: standard deviation. * Student’s *t*-test for paired samples. CI: confidence interval. ** Agreement by the Bland and Altman technique. Pan: panoramic radiography. Peri: periapical radiography. T13/23: tooth 13/23.

**Table 5 diagnostics-14-00749-t005:** Comparison of actual age and estimated age by various formulas for premolars (*n* = 247).

Equation	Mean ± SD (Years)	Difference in Years	*p* *	CI (95%) **
Cameriere et al. [2]				
Pan T35/45	40.3 ± 7.8	+4.1	<0.001	−16.5 to 24.7
Peri T35/45	42.3 ± 7.6	+6.1	<0.001	−14.7 to 26.9
Lee et al. [12]				
Pan T35/45	48.7 ± 10.5	+12.5	<0.001	−9.1 to 34.1
Anastácio et al. [13]				
Pan T35/45	24.2 ± 1.8	−12.1	< 0.001	−11.0 to 35.2

SD: standard deviation. * Student´s *t*-test for paired samples. CI: confidence interval. ** Agreement by the Bland and Altman technique. Pan: panoramic radiography. Peri: periapical radiography. T35/45: tooth 35/45.

**Table 6 diagnostics-14-00749-t006:** Pixel values of pulp/tooth ratio obtained in teeth 13/23 and 35/45 in panoramic and periapical radiographs.

Radiography/Tooth	n	Mean	SD	Min	Max
Pan 13/23	247	0.1025	0.0236	0.0295	0.1716
Peri 13/23	247	0.1064	0.0215	0.0429	0.1598
Pan 35/45	247	0.0992	0.0214	0.0350	0.1537
Peri 35/45	247	0.0937	0.0210	0.0247	0.1535

SD: standard deviation. Min: minimum. Max: maximum. Pan: panoramic radiography. Peri: periapical radiography.

**Table 7 diagnostics-14-00749-t007:** Equations obtained by applying the linear regression model in the Brazilian subpopulation.

Radiography/Tooth	Brazilian Subpopulation Equation	R^2^	MAE	SD
Pan T13/23	Age = 68.348 − 313.070 × R13/23	0.34	106.47	10.32
Peri T13/23	Age = 73.245 − 347.648 × R13/23	0.35	105.20	10.26
Pan T45/35	Age = 69.447 − 334.606 × R35/45	0.32	109.53	10.47
Peri T45/35	Age = 67.067 − 328.886 × R35/45	0.30	113.34	10.65

R^2^: determination coefficient. MAE: mean absolute error. SD: standard deviation. Pan: panoramic radiography. T13/23: tooth 13/23; Peri: periapical radiography. T35/45: tooth 35/45.

## Data Availability

The data are available in this article.

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
