# Peer review of "Age Estimation in Brazilian Adults Using the Pulp/Tooth Ratio of the Maxillary Canine and Mandibular Second Premolar"

_diagnostics, 2024, doi:10.3390/diagnostics14070749_

Round 1

Reviewer 1 Report

Comments and Suggestions for Authors

Dear authors,

Congratulations on the job you have done and presented in this manuscript. I believe that your work is significant to the field and is of high interest for the general reader. There are some modifications required prior to considerations for publication, especially in the material and method section. Please see the attachment.

Author Response

I appreciate all the suggestions. Changes made to the text were highlighted in red.

Reviewers’ comments

Answers

Reviewer 1

1. Title is very long, please try to rephrase to make it more understandable for the general reader (you can remove panoramic and periapical).

The title was changed according to the reviewer's suggestions.

2. Abstract is very well structured, meets all the required standards.

Thanks for the comment.

3. Authors must state the hypothesis of the present research before the aim.

A null hypothesis was inserted at the end of the introduction.

4. Did the authors calculated the sample size required to perform this study? How was sample size determined?

Information about the sample size calculation was inserted in the text.

5. Figure 2?

The caption has been corrected.

6. (C) pulp area is barely visible in the apical third, so the measurement performed was very subjective considering the poor visibility in that area. How did the authors avoid errors during measurements?

Cameriere’s methodology recommends using a minimum of 20 points for outlining the tooth and 10 points for outlining the pulp. For the execution of this study, a pen and digitizing table were chosen to ensure ease, precision, and standardization of markings.

All measurements were conducted by a single observer, a dental radiology specialist with 30 years of professional experience, who underwent prior calibration.

The consistently high ICC values exceeding 0.99% across all measurements underscore the reproducibility and standardization achieved in this study.

7. Please move this section at the end of materials and methods section, because this is part of your methodology.

I'm sorry, but the text is already in the final portion of the methodology section.

8. Results are clearly presented and comprehensive, no modifications required.

Thanks for the comment.

9. Please state what are the limitations and strengths of the present study.

Information about the limitations and strengths of the study was included in the discussion.

10. It will look much better if the authors present the conclusions into a single paragraph.

The conclusion was modified according to the reviewer's suggestions.

Reviewer 2 Report

Comments and Suggestions for Authors

This is a very well written study, with detailed background section and thorough discussion appropriately documented with citations, that, despite the not statistically significant results, represents a role model paper regarding the study design and presentation of methods and results, and to this reviewer opinion merits consideration for publication. Some minor revisions are recommended:

1) Some results sections could be summarized, e.g., in lines 234-272, the same sentence "The dispersion diagram of the pulp/tooth ratio...when 261 genders are evaluated." is repeated 4 times for the different tooth category or x-ray type. This information should be summarized in one sentence.

2) The study conclusions should be focused on the little agreement between the actual age and the estimated age. Comments about "The equations employed allowed for estimating the age" (line 26) should be avoided due to the little aggreement. Similarly "New equations were developed" (line 404) should be changed to "New equations should be developed".

3) Line 32: delete remains

4) Line 34, please change to "Accurate determination of the chronological age"

5) Line 67 "healthy maxillary canine": please explain what do you mean "healthy"? e.g., withought caries?

Author Response

I appreciate all the suggestions. Changes made to the text were highlighted in red.

Reviewers’ comments

Answers

Reviewer 2

This is a very well written study, with detailed background section and thorough discussion appropriately documented with citations, that, despite the not statistically significant results, represents a role model paper regarding the study design and presentation of methods and results, and to this reviewer opinion merits consideration for publication. Some minor revisions are recommended:

Thanks for the comment.

1. Some results sections could be summarized, e.g., in lines 234-272, the same sentence "The dispersion diagram of the pulp/tooth ratio...when 261 genders are evaluated." is repeated 4 times for the different tooth category or x-ray type. This information should be summarized in one sentence.

The text was modified according to the reviewer’s suggestions.

2. The study conclusions should be focused on the little agreement between the actual age and the estimated age. Comments about "The equations employed allowed for estimating the age" (line 26) should be avoided due to the little agreement. Similarly, "New equations were developed" (line 404) should be changed to "New equations should be developed".

The conclusion presented in the abstract (lines 25 and 26) was changed following the reviewer's suggestions.

Regarding the second part of the conclusion (lines 410 to 412), we believe that no changes are necessary. The present study aimed to compare the effectiveness of different equations in estimating the age of an adult Brazilian subpopulation by calculating the pulp/tooth ratio of the maxillary canine and mandibular second premolar in panoramic and periapical radiographs, and to develop equations for estimating age based on values obtained from this sample.

“New equations were developed…” addresses precisely this second objective.

At the end of the discussion (lines 404 to 407), the future perspectives opened by this research are presented. They include reasoning for the need to develop new equations.

3. Line 32: delete remains.

The text was modified according to the reviewer’s suggestions.

4. Line 34, please change to "Accurate determination of the chronological age".

The text was modified according to the reviewer’s suggestions.

5. Line 67 "healthy maxillary canine": please explain what do you mean "healthy"? e.g., without caries?

Healthy was used to describe that the included teeth that did not have restorative treatment were free from cavities (dental caries).

Round 2

Reviewer 1 Report

Comments and Suggestions for Authors

Dear authors,

I believe that you took advantage of the suggestions and improved the manuscript. Since, I have no further comments, I will suggest publication of your article. Congratulations!